

# A description and sensitivity analysis of the ArchMatNet agent-based model

Robert J. Bischoff[1,*] and Cecilia Padilla-Iglesias[2,*]

[1] School of Human Evolution and Social Change, Arizona State University, Tempe, AZ, United States of America
[2] Department of Evolutionary Anthropology, University of Zurich, Zurich, Switzerland
[*] These authors contributed equally to this work.

## ABSTRACT

Archaeologists cannot observe face-to-face interactions in the past, yet methods derived from the analyses of social networks are often used to make inferences about patterns of past social interactions using material cultural remains as a proxy. We created the ArchMatNet agent-based model to explore the relationship between networks built from archaeological material and the past social networks that generated them. It was designed as an abstract model representing a wide variety of social systems and their dynamics: from hunter-gatherer groups to small-scale horticulturalists. The model is highly flexible, allowing agents to engage in a variety of activities (*e.g.*, group hunting, visiting, trading, cultural transmission, migration, seasonal aggregations, *etc.*), and includes several parameters that can be adjusted to represent the social, demographic and historical dynamics of interest. This article examines how sensitive the model is to changes in these various parameters, primarily by relying on the one-factor-at-a-time (OFAT) approach to sensitivity analysis. Our purpose is for this sensitivity analyses to serve as a guide for users of the model containing information on how the model works, the types of agents and variables included, how parameters interact with one another, the model outputs, and how to make informed choices on parameter values.

# INTRODUCTION

Understanding social interactions in the past is vital for gaining insights into how social structures, patterns of social organization and human adaptations to environmental and social changes have changed throughout human evolution in different places. It is also essential for determining the extent of the diversity in human behaviour. Social network analyses present an important tool for studying human interaction patterns at both the individual or community levels, and therefore gain such insights into the nature of social relationships as well as social structures (*Apicella et al., 2012*; *Mace et al., 2018*; *Von Rueden et al., 2019*; *Padilla-Iglesias & Kramer, 2021*). Such social networks are constructed on the basis of particular kinds of interactions between individuals (*e.g.*, proximity, joint engagement in particular activities, economic transfers, *etc.*).

Corresponding authors
Robert J. Bischoff, rbischoff@asu.edu
Cecilia Padilla-Iglesias, cecilia.padillaiglesias@uzh.ch

Archaeologists cannot observe face-to-face interactions in the past, yet social network methods are often used to make inferences about patterns of past social interactions using material cultural remains as a proxy (*Mills et al., 2013*; *Peeples, 2018*). The ArchMatNet agent-based model (ABM) was designed to investigate archaeologists' ability to reconstruct prehistoric social networks from networks of material culture under different conditions. For example, the presence of similar ceramic styles at different site locations has often been used to create a network tie between these locations (*e.g.*, *Mills et al., 2013*; *Borck et al., 2015*; *Peeples, 2018*; *Birch & Hart, 2018*; *Lulewicz, 2019*). The question remains: How do networks created in this way compare to the networks of social relations that existed in the past, which created the archaeological record? This is the question that ArchMatNet is designed to evaluate. Given that cultural transmission and evolution are affected by multiple interacting phenomena, the purpose of our model is to evaluate how different factors relating to the size and structure of social groups, local environments, individual learning strategies or properties of the cultural traits themselves may affect the relationship between social and material cultural networks. These relationships may differ based on the scale, social structure, and material culture of a society. Our model, though generalized, is designed to represent hunter gatherers and small-scale horticulturalists. We model various activities, such as cultural transmission, visiting, group hunting, trading, and migration. Numerous parameters, including population size, can be adjusted to fit many different scenarios. Given the complexity of human behavior, the model cannot fundamentally prove the relationship between material culture and social networks. Rather, the model is a theoretical test of this relationship under known, controlled conditions.

However, before we can use ArchMatNet to explore our fundamental questions, we must first ask questions regarding the workings of the model itself. This is the primary objective of this article. ABMs function as "behavioral laboratories" that allow us to control the conditions in the experiment (*Premo, 2006*; *Graham & Weingart, 2015*). Hence, to adequately understand the results of our model, we must determine how our starting parameters interact with each other and affect the model outputs.

Two of the primary questions when designing and validating an ABM are: "What parameters should be included?" and "What should the values for such parameters be?" Sensitivity analyses are necessary when precise values are not known for a parameter (*Hamby, 1994*; *Saltelli et al., 2004*; *Niida, Hasegawa & Miyano, 2019*). This is often the case when modeling societies. Sensitivity analyses are essential tools to confirm the robustness of the results by demonstrating how outputs respond to a range of parameter values, and provide insights into the dynamics of the system (*Leamer, 1983*; *Leamer, 2010*; *Axtell, 1999*; *Sargent, 2013*; *Romanowska, 2015*; *Ten Broeke, Van Voorn & Ligtenberg, 2016*; *Brouwer Burg, Peeters & Lovis, 2016*; *Niida, Hasegawa & Miyano, 2019*; *Kanters, Brughmans & Romanowska, 2021*). They also can reduce the number of model runs by eliminating parameters that do not impact results (*Romanowska, 2015*). Documenting emergent phenomena is particularly important in agent based modeling, and sensitivity analyses need to identify these patterns and how robust they are (*Ligmann-Zielinska et al., 2014*; *Ten Broeke, Van Voorn & Ligtenberg, 2016*). Due to the large number of parameters in our model and particularly due to the lack of a specific target metric, we rely primarily

on a one-factor-at-a-time (OFAT) approach (*Czitrom, 1999*; *Ten Broeke, Van Voorn & Ligtenberg, 2016*).

This article is a description of the parameters we chose and a sensitivity analysis that explores how the parameters relate to each other and affect the model. It is not an exhaustive description of the model; which can be found in the ODD (Overview, Design concepts, and Details–see *Grimm et al. (2010)*). This model was written using NetLogo 6.2.2 (*Wilensky, 1999*) and is publicly available–see Data Availability Statement for link to ODD and ArchMatNet model.

## MODEL DESCRIPTION

We first provide a narration of the model, which we follow with a more technical description. The model is initiated with a variable number of bands scattered around a landscape with either separate environments or a mutating environment. These bands represent either hunter-gatherer bands or multi-settlement communities of small-scale horticulturalists. Each band has three camps and each camp has a variable, but identical number of people that live in the camp. Each time step of the model provides an opportunity for each person to visit allies, go hunting together, migrate, create new styles of objects, learn from each other, trade objects, make new objects, and break objects. At a specified interval, everyone belonging to the same band gathers together for a time to visit, learn from each other, and trade. These activities continue until the end of the simulation. Each interaction is recorded and this forms one of the primary outputs–the social network between agents. Objects dropped at camps are recorded in the recorded in the camp assemblage and this forms the archaeological record–the second primary output. The ultimate purpose of these outputs is to compare the social networks with networks created from the material culture.

A simplified flowchart describing how the model procedures are called is shown in Fig. 1. The *go* procedure is run once every time step. Figure 2 shows the model interface. The model can be opened with Netlogo and controlled through the interface. The model is initialized with the *setup* button and runs using the *go* button. It can also be run one step at a time using the *go once* button. A number of sliders and switches control the parameters. Other switches and buttons control the outputs. The random seed used in the model can also be held constant to reproduce results. The interface provides real-time feedback on the number of distinct traits in the model, the point prestige, and the Jaccard distance between camps (see sections below for a description of these metrics).

### Model environment

Our model contains three types of agents: people, camps (comprising multiple people), and bands (comprising multiple camps). There are two types of camps: Hunter-gatherer residential camps where individuals live (*Binford, 1980*; *Kelly, 1983*); and sites where seasonal aggregations take place (a common practice among hunter-gatherers; *Rorabaugh, 2019*; *Lieberman et al., 1993*). Because the camps are stationary, they can also represent villages in small-scale horticultural societies. The model world is a square grid divided into band territories, as well as environmental zones (analogous to different ecosystems).
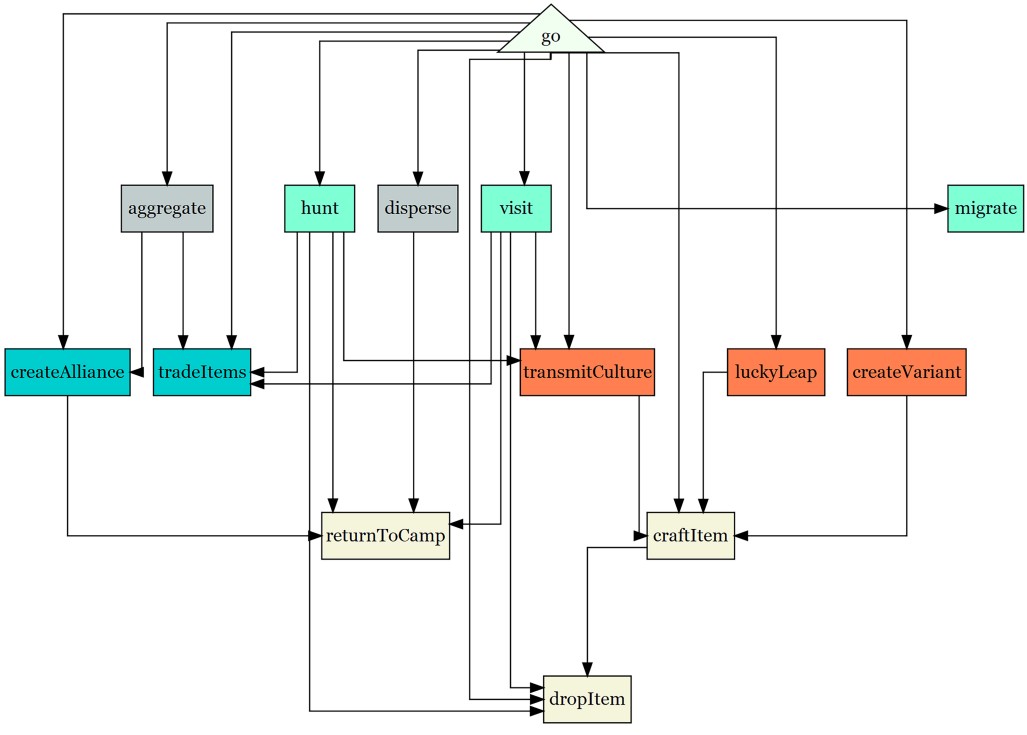

**Figure 1  Flowchart showing the most important procedures in ArchMatNet.** The Go procedure is the main procedure that begins each time step. This chart shows which procedures are called by the Go procedure and what procedures are called by other procedures.

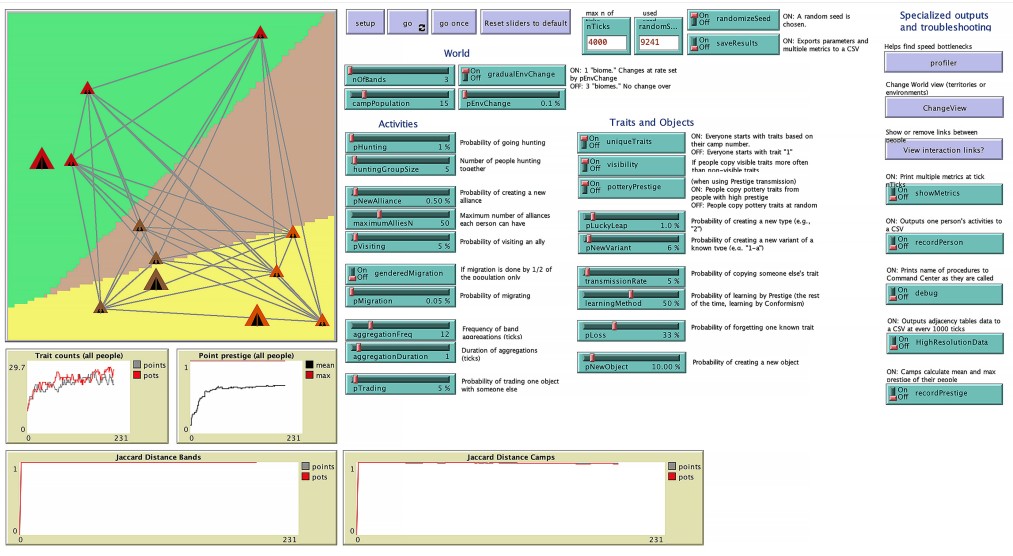

**Figure 2  NetLogo interface showing camps, bands, and people with interaction links between people (people are located within camps) as well as adjustable parameters from the model.**

Alternatively, we also include a possibility for the environment to be spatially homogeneous yet changing with a specified probability. Each time step represents a month.

People meet one another through hunting, visiting and engaging in seasonal aggregations. Hunting is done exclusively with people from one's own band, but visiting is done between allies, which can be of other bands. In the ethnographic literature, it is well-documented that hunter-gatherers tend to travel further distances to visit acquaintances and family members than to carry out subsistence activities (*Wiessner, 1977*; *Hewlett, Van de Koppel & Cavalli-Sforza, 1982*; *Hill et al., 2014*). Similarly, individuals also aggregate with members of their own band (*Bahuchet, 1991*; *Lieberman et al., 1993*; *Lewis, 2015*).

In addition to temporary moves, we also include migration. Migration implies that an individual changes their residence camp permanently. In the literature, one of the key events that marks the permanent migration of people from small-scale societies to a different residential location is the post-marital change of residence (*Kelly, 2013*; *Dyble et al., 2015*; *Moravec et al., 2018*). Given that societies differ with regards to whether only members of one gender or both tend to change residence after marriage, in our model we include also the possibility of gendered migration. That is, of only allowing members of one sex to change their residence camp. This corresponds to patrilocality/matrilocality.

People have cultural inventories of up to 10 pots and 10 points. These two types of objects represent cultural domains that have been hypothesized to be subject to different cultural evolutionary pressures (and therefore respond differently to changes in ecological or social dynamics). This is because they serve different functions. Points, and other subsistence tools must be adaptive for the extraction of available resources in specific environments, which may potentially promote high-fidelity copying to prevent "maladaptive errors" and bias the transmission of certain functional (*i.e.,* more efficient) variants (*Arthur, 2009*; *Rogers, Feldman & Ehrlich, 2009*). On the other hand, pots, and other stylistic traits are not necessarily subject to ecological pressures, and instead, their transmission might be affected by other factors such as aesthetics, the desire to converge with others or convey membership to a particular group (*Da Silva & Tehrani, 2016*). Precisely because of that, we have also implemented two types of cultural transmission: prestige-based and conformist-based. During prestige-based cultural transmission, people identify within their camp location the most prestigious individual, and copies one of their point or pot traits. For points, people acquire prestige depending on the number of successful hunts they have achieved. For pots, prestige can either be deactivated or assigned in a random fashion at the start of the model. When deactivated, a person is chosen at random to be have the highest prestige. We include this option, to include the possibility of popular people's behaviour to be copied even when the behaviour itself does not have a selective advantage. This is analogous to people copying fashion trends of important individuals. During prestige-based cultural transmission, people identify within their current location the most prestigious individual, and copy one of their point or pot traits. During conformist-based cultural transmission, people identify within their current location the most common trait for a particular tool type, and adopt it.

Objects from those inventories are created based on traits in the agent's repertoire. Main trait types are represented as integers. People can invent either a new trait type or new

variants of a type they know (see *Kolodny, Creanza & Feldman, 2015*; *Creanza, Kolodny & Feldman, 2017* for a similar approach). Variants are represented as discrete letters (a to o only). Each point trait is adaptive in the environment where it was innovated. For example, a point trait created by an agent in one environment will be adapted to that environment and not necessarily in another environment.

The model has an option to include trait visibility. Visibility is essential for understanding material styles and the underlying social behaviors generating them (*Wobst, 1977*; *Carr, 1995a*). Some traits, like the size of a pot or large designs, are easy to see. These traits are high-visibility traits. Other traits, like the temper used in pottery, are difficult to see. These are low-visibility traits. Low-visibility traits generally spread through observation of production or active teaching, whereas high-visibility traits can easily be observed in the finished product. When visibility is switched on in the model, high-visibility traits are preferentially learned over low-visibility traits. Traits are given an even chance of being high-visibility or low-visibility, but learning procedures select high-visibility traits most of the time in accordance with relevant literature (*e.g.*, *Carr, 1995b*; *Carr, 1995a*; *Clark, 2001*).

## Overview of model dynamics

At each time-step, the dynamics of the model proceed as follows:

1. ENVIRONMENTAL CHANGE: First, if the probabilistic environmental change is specified, the model will randomly throw a die to see if the environment will change. If it does, a new environment will be set. People remove any alliances they have made that exceed the maximum parameter (agents only visit people they have an alliance with and the number of alliances is controlled).

2. INNOVATION: People then have the opportunity to invent new tool types and new variants and create one pot and point each. A person may create a new pot and/or point each time step. Each new object is created based on a randomly selected trait from the person's trait list. Each object also records the time step when it was created for use in subsequent analyses (as this is analogous to an archaeological date).

3. ACTIVITIES: The model then checks if it is time to aggregate based on a user-specified probability. If it is, people move to their respective aggregation camps. People aggregate for a specified duration, and then go back to their own camp. Whilst aggregating, people are recorded as "visiting" those who they are aggregated with. During aggregations, people can also go hunting with a certain probability.

When people are not aggregating, they can take part in several activities with user-specified probabilities: They can create a new alliance with members of other camps, they can go visit their alliances, or they can go hunting. When hunting, people gather other people from their band and go to a different location together. When the hunt occurs during an aggregation, the hunting site is near the patches of the aggregation sites. When the hunt occurs at other times, the hunting site is simply within the band territory. On a hunt, the hunters have different probabilities to drop a point, trade points with one another, and transmit trait knowledge to one another. Hunters keep track of how many times they hunted together. When the hunt is over, everyone returns to their own camp.

When visiting, one person chooses one ally to go visit at random. They move to that ally's camp and visit with all the allies present there. The people who are involved in the visit keep track of how many times they visited one another. The people who participate in the visit have different probabilities to drop one pot, drop one point, trade objects with one another, and transmit trait knowledge to one another. When those activities are done, the visitor returns to their own camp.

5. CULTURAL LOSS: All people have a certain probability to forget one pot or point trait.

6. CULTURAL TRANSMISSION: All people have the possibility to exchange the knowledge of traits, and to trade objects from their inventories with one another.

7. GENERATING THE ARCHAEOLOGICAL RECORD: People may drop a pot of a point depending on a set probability. This is what generates the archaeological record in the model.

8. MIGRATION: After the first 1000 time steps, people may migrate every time step based on a probability set by the user. When migration occurs, the person who is migrating chooses another person to switch residential camps. There is also the possibility for gendered migration, whereby only people belonging to one gender can switch camps.

## Model outputs

The model has a number of outputs and several ways to access them. The primary output is the raw data. The *saveResults* option produces four CSV files. These files include the virtual artifact assemblages for each camp, the number of interactions between each person, the metadata for each person (which camp and band they belong to), and a list of all the variables used to run the experiment. There is one drawback for this raw data, as the interaction links can provide inaccurate results when summarized by camp or band due to migration. An aggregating function can be used to produce adjacency tables that include accurate interaction counts between bands and camps. A *HighResolutionData* option can be used to export these adjacency tables, as well as adjacency tables for the pots and points that show the weighted Jaccard distance between each pair of assemblages at every 1000 ticks. The weighted Jaccard distance provides a value between zero (most similar) and one (most distant) and is calculated as follows:

$$J_w(x,y) = \frac{\Sigma_i min(x_i, y_i)}{\Sigma_i max(x_i, y_i)}$$

These adjacency tables can also be returned when running behavioral experiments in NetLogo. One additional CSV file can be generated, although it is primarily for debugging purposes. The *recordPerson* option selects one agent and records each activity and the tick the activity occurred. It outputs this information in a CSV, which can be useful for closely tracking the frequency and timing of activities.

Besides the raw output, there are several other metrics the model can calculate. The *showMetrics* option produces a number of statistics, shown in Table 1. There are also functions available that can be called from the NetLogo Command Center or Behavior Space. The Jaccard distance for the number of traits held between camps and bands can be computed. Two additional network comparison methods are included: One is an absolute

**Table 1  Metrics Included in the ArchMatNet Output.**

| Metric | Scope | Description |
| --- | --- | --- |
| Assemblage size | Pots, points | Total number of artifacts deposited in camp |
| Interaction frequency | Hunting, learning, trading, visiting | Total number of interactions |
| BR distance | Pots, points | Brainerd-Robinson distance between each camp and band using the assemblage |
| Prestige | Points | Mean point prestige |
| Modularity | Pots, points | Modularity metric which evaluates if there is more between-band links than within-band links, based on the Jaccard similarity of traits known by people |
| Network metric | Pots, points | Difference between the clustering coefficient of the network created by people who interacted with one another and the clustering coefficient of the network created by similarities in objects |
| PhiST | Pots, points | PhiST measure to see if variation in known traits is structured around bands |
| Number of traits | Pots, points | Total number of distinct traits |

matrix comparison that binarizes the adjacency matrices using the median value and determines the proportion of edges that are shared between the networks. The other metric requires the *sr* extension enabled in NetLogo, as it relies on the "multinet" (*Magnani, Rossi & Vega, 2020*) package in R (*R Core Team, 2022*) to calculate the Pearson correlation coefficient between the network edges. These outputs provide several ways to evaluate the output with or without further analysis.

## SENSITIVITY ANALYSES

These sensitivity analyses use only a few of the available metrics. Our interest here is not to analyze the results of the model or contextualize the model outcomes in terms of their archaeological relevance. Instead, we wish to determine how different parts of the model affect its outputs (*i.e.,* metrics). The analyses were conducted by running individual *BehaviorSpace* experiments in NetLogo for each parameter. The number of runs depended on how many values were chosen for each parameter and how much variability was shown by each parameter. In total, 2,988 runs of the model were completed for the sensitivity analyses with between 70 and 210 runs for each of the 24 parameters. Spearman's rank correlations, $r^2$ values, *p*-values, and 95% confidence intervals were calculated for each parameter and 13 metrics using R (*R Core Team, 2022*). The R script, full results, and related plots are available at OSF: Bischoff, Robert, Cecilia Padilla-Iglesias, and Claudine Gravel-Miguel. 2023. "Supplemental Material for A Description and Sensitivity Analysis of the ArchMatNet Agent-Based Model." OSF. April 11. doi: 10.17605/OSF.IO/C6M5T.

Figure 3 shows the metrics and parameters used in the OFAT (one factor at a time) analysis. We found a number of strong correlations between the model parameters and the different outputs produced. We divide the discussion of the sensitivity analyses into three sections: parameters related to the environment, parameters related to sociality, and parameters related to cultural behavior and material culture. We discuss how the

| | getJaccardBandPots | getJaccardBandPoints | timer | nTraitsPot | nTraitsPoint | interactionFreqHunting | interactionFreqVisiting | interactionFreqTrading | interactionFreqLearning | assemblageSizePoint | assemblageSizePot |
|---|---|---|---|---|---|---|---|---|---|---|---|
| pTrading | 0 | 0 | 0.3 | 0.1 | 0 | 0 | 0 | 1 | 0 | 0 | -0.1 |
| maximumAlliesN | -0.2 | 0 | 0 | -0.1 | -0.1 | 0 | 0.4 | 0.4 | 0.3 | 0.1 | 0.1 |
| pHunting | -0.1 | 0.3 | -0.3 | -0.1 | -0.1 | 1 | 0 | 0.6 | 0.6 | -0.3 | 0 |
| huntingGroupSize | 0 | 0.1 | 0.1 | 0 | -0.1 | 1 | 0 | 0.3 | 0.3 | -0.2 | 0 |
| pLoss | 0.4 | 0.3 | -0.5 | -1 | -1 | 0 | 0 | 0 | 0 | 0.1 | 0 |
| pMigration | -0.9 | -0.9 | -0.2 | 0.1 | -0.1 | 0.3 | -0.1 | -0.1 | 0 | 0.2 | 0.2 |
| learningMethod | -0.1 | -0.3 | 0.2 | 0 | 0 | 0 | 0 | 0 | -0.9 | 0 | -0.1 |
| potteryPrestige | -0.1 | -0.1 | -0.2 | 0 | 0 | 0 | 0 | 0 | 0 | 0 | 0 |
| gradualEnvChange | 0 | 0 | -0.3 | 0 | -0.1 | 0 | 0 | 0 | 0 | 0 | 0 |
| pNewAlliance | 0.1 | -0.1 | -0.1 | 0.1 | -0.1 | 0.1 | -0.3 | -0.3 | -0.1 | 0 | 0 |
| pEnvChange | 0 | 0 | 0.3 | 0.1 | 0 | 0 | 0 | 0 | 0 | 0 | 0 |
| aggregationDuration | -0.2 | -0.1 | 0 | -0.1 | -0.1 | 0.1 | -0.4 | -0.4 | 0.4 | -0.1 | 0 |
| aggregationFreq | 0 | 0.1 | 0 | 0.1 | 0.1 | 0.4 | -0.9 | -0.4 | -0.3 | 0 | 0 |
| pNewObject | 0 | 0 | -0.3 | 0.1 | -0.1 | 0 | 0 | 0.3 | 0 | 0.9 | 0.9 |
| uniqueTraits | 0.9 | 0.9 | -0.1 | 0 | 0 | 0 | 0 | -0.1 | 0 | 0 | 0 |
| genderedMigration | 0.6 | 0.7 | 0.2 | -0.1 | 0.1 | 0 | 0.4 | 0.3 | 0 | 0.1 | 0 |
| pNewVariant | 0.7 | 0.6 | 0.2 | 0.8 | 0.8 | 0 | 0 | 0.1 | 0 | 0.3 | 0 |
| pLuckyLeap | 0.9 | 0.9 | 0.2 | 0.5 | 0.4 | 0 | 0 | 0 | 0 | 0.3 | 0.3 |
| visibility | 0.5 | -0.8 | 0.2 | -0.7 | -0.7 | 0 | -0.1 | 0 | -0.9 | -0.8 | -0.8 |
| transmissionRate | -0.1 | 0.7 | 0.5 | 0.6 | 0.3 | 0.1 | 0 | 0.5 | 1 | 0.7 | 0.7 |
| pVisiting | -0.1 | 0.3 | 0.4 | 0.1 | 0.1 | -0.1 | 0.8 | 0.9 | 0.9 | 0.5 | 0.6 |
| nTicks | -0.1 | 0.5 | 0.8 | 0.1 | -0.1 | 1 | 1 | 1 | 1 | 1 | 1 |
| nOfBands | 0.9 | 0.9 | 0.9 | 0.9 | 0.9 | 0.9 | 0.9 | 0.9 | 0.9 | 0.9 | 0.9 |
| campPopulation | -0.9 | -0.9 | 0.7 | 0.9 | 0.9 | 0.9 | 0.9 | 0.9 | 0.9 | 0.9 | 0.9 |

**Figure 3  Correlation matrix showing the calibration variables and metrics used in this analysis.** Red is positive correlation and blue is negative correlation.

parameters on each of those sections relate to one another and affect our model's simulated "archaeological record".

## Environmental factors

The environmental factors relate to the duration of the model and the ways in which we modeled the environment. These parameters are:

- nTicks
- pEnvChange

As expected, the longer we ran our model, the greater the number of pots and points produced (as they are produced at regular frequencies), and the greater number of interactions between agents (as these also occur at regular frequencies). However, running our model for longer did not affect the overall cultural differentiation between assemblages, meaning that the dynamics of the model, and model outputs did not seem to be altered by running our model for more than 2,000 time steps.

Surprisingly, all other things being equal, the frequency of environmental change, *pEnvChange*, did not significantly affect any of the model outputs. That is, it did not alter the overall cultural differentiation between assemblages, or the frequency of interactions between model agents for any of the activities recorded.

## Social factors

Social factors are parameters that control social behaviors relating to population size and the frequencies of various activities. These parameters are:

- aggregationDuration
- aggregationFreq
- campPopulation
- genderedMigration
- huntingGroupSize
- maximumAlliesN
- pHunting
- pMigration
- pNewAlliance
- pTrading
- pVisiting

Similarly to what we observed when running the model for longer, increasing the number of agents in the model through increasing camp populations, or increasing the number of allies people were allowed to have led to more pots and points being produced (see Fig. 4), and a greater number of interactions between agents. However, having smaller camp populations, led to a greater cultural differentiation between assemblages for both pots and points (see Fig. 5).

Increasing the frequency and duration of seasonal aggregations did not affect any of the model outcomes with the exception of the frequency of visit interactions, given that whilst aggregated, the counter for visits between all agents aggregated at the same site increases.

The probability of migration events negatively affected the assemblage size of pots and points (see Fig. 6), as well as the interaction frequency of visiting and trading. Moreover, reduced migration resulted in assemblages being more culturally distinct from one another (*i.e.,* increased the Jaccard distances between assemblages for both pots and points). Interestingly, having only one gender allowed to migrate resulted in a similar effect, even at similar migration rates (see Fig. 7).

The probability of visiting other agents had an effect on the assemblage size of pots, but not of points. At very high visiting probabilities, we also observed reduced Jaccard distances

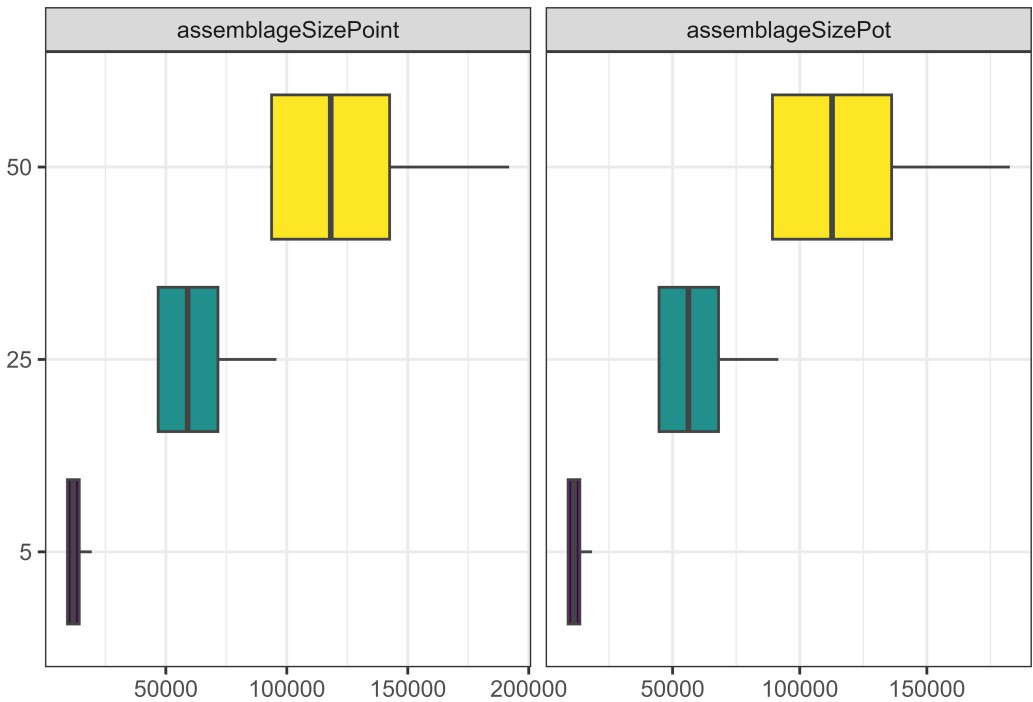

**Figure 4** **Boxplot demonstrating the correlation between the camp population parameter and the assemblage sizes.** The x axis is the size of assemblages. The y axis is the number of people starting in each camp.

between the pot assemblages of camps. This was not the case when considering differences between the pot assemblages of bands or any of the point assemblages. The probability of visits also affected the frequency of learning and trading interactions.

The hunting probability and hunting group size also affected the frequency of hunts as well as the frequency of trading and learning events. However, more frequent hunts also reduced the cultural differentiation between the pot assemblages of camps. The probability of creating new alliances did not affect any of the model outputs.

## Cultural/Material factors

The cultural/material factors relate to the innovation and transmission of traits, as well as traits inherent to the material objects. These parameters are:

- learningMethod
- pLuckyLeap
- pNewVariant
- pNewObject
- potteryPrestige
- transmissionRate
- uniqueTraits
- visibility
- pLoss

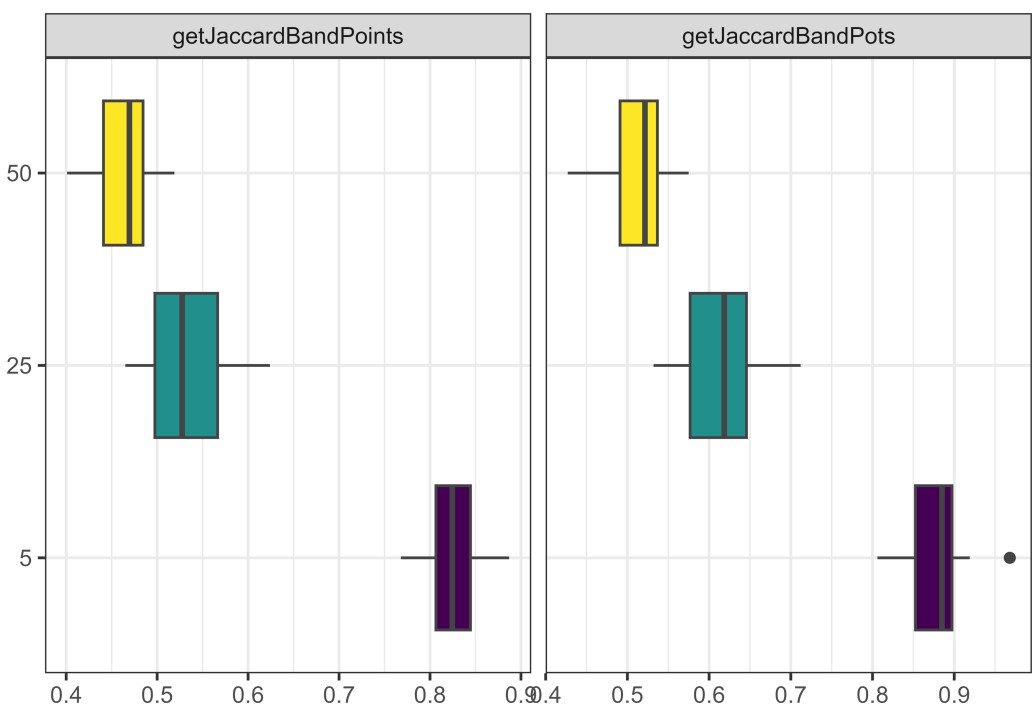

**Figure 5** **Boxplot demonstrating the correlation between the camp population parameter and the Jaccard distances.** The x axis is the Jaccard distance between bands' point and pot assemblages. The y axis is the number of people starting in each camp.

Out of these parameters, the one that had the greatest effect on the model dynamics was whether each camp started with unique traits. Figure 8 demonstrates how the Jaccard distance between bands correlates with starting with unique traits. None of the other parameters discussed in this section had a stronger correlation (0.87) with the Jaccard distance between band, but this metric also had no correlation with any other metric.

Another important parameter was the probability of losing a trait (*i.e.,* forgetting a learned or innovated trait). As Fig. 9 shows, the correlation is not straightforward. What is clear is that this parameter is an important part of creating cultural distance as runs of the model with zero loss consistently resulted in small Jaccard distances between bands.

The lucky leap (*i.e.,* invention of a new trait) only had a major affect on one metric: more innovation led to greater Jaccard distances between bands (see Fig. 10). A related parameter, the probability of creating a new trait (*pNewVariant*), primarily affected the number of traits found in the model but also had a significant effect on the Jaccard distance between bands with more traits resulting in a greater distance between bands.

The probability of creating a new object led to larger assemblages and slowed down the model. It also increased the frequency of trading, as agents create a new object when they learn a new trait. We consider this a natural part of learning. This increases the number of objects available to trade. Sometimes trading does not happen because there are no objects in a person's inventory.

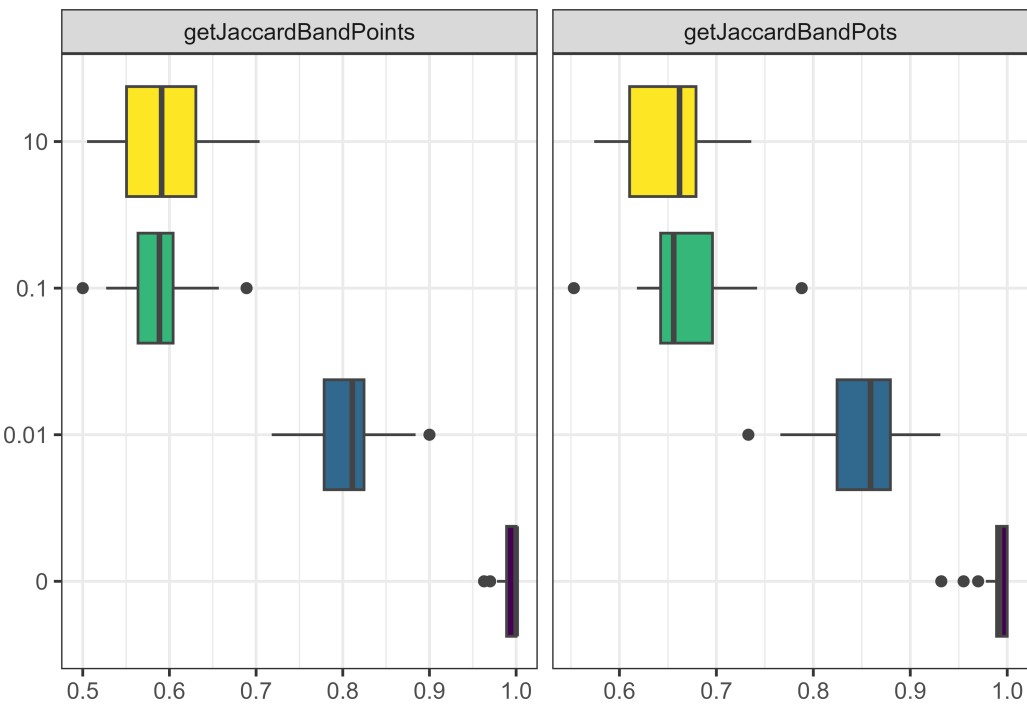

**Figure 6** **Boxplot demonstrating the correlation between the probability of migration parameter and the Jaccard distances.** The x axis is the Jaccard distance between bands' point and pot assemblages. The y axis is the percentage probability of a person migrating each time step.

The learning method significantly affected one metric: the frequency of learning. Conformism involves more people, thus there are more learning interactions. The Jaccard distance between bands was moderately affected by the learning method. Prestige learning increased the distance between bands, and the effect was stronger for pots than points.

Closely related to the learning method is the transmission rate, but this parameter had some of the more interesting results. Increased learning decreased the Jaccard distance between bands when measured by pot traits, but increased the Jaccard distance between band when measured by point traits. Figure 11 shows that the Jaccard distance for pots with low transmission is greater than that for no transmission, but decreases as the transmission rate increases. This is expected as more learning allows traits to diffuse among bands. However, the functional value of points allows the Jaccard distance to continue to increase as the transmission rate increases. More learning was also strongly correlated with the size of the assemblages (as new objects are created when learned). More objects available also increased the frequency of trading. The number of traits was also correlated with learning, as the trait is less likely to be lost, although pot traits were more strongly correlated than point traits.

The parameter most directly related to the object itself is visibility. Once again, pots and points differed in their Jaccard difference for this parameter, but in this case points were negatively correlated. When visibility was on, point traits were more similar between. Whereas pot traits were more distinct between bands when visibility was on. High visibility

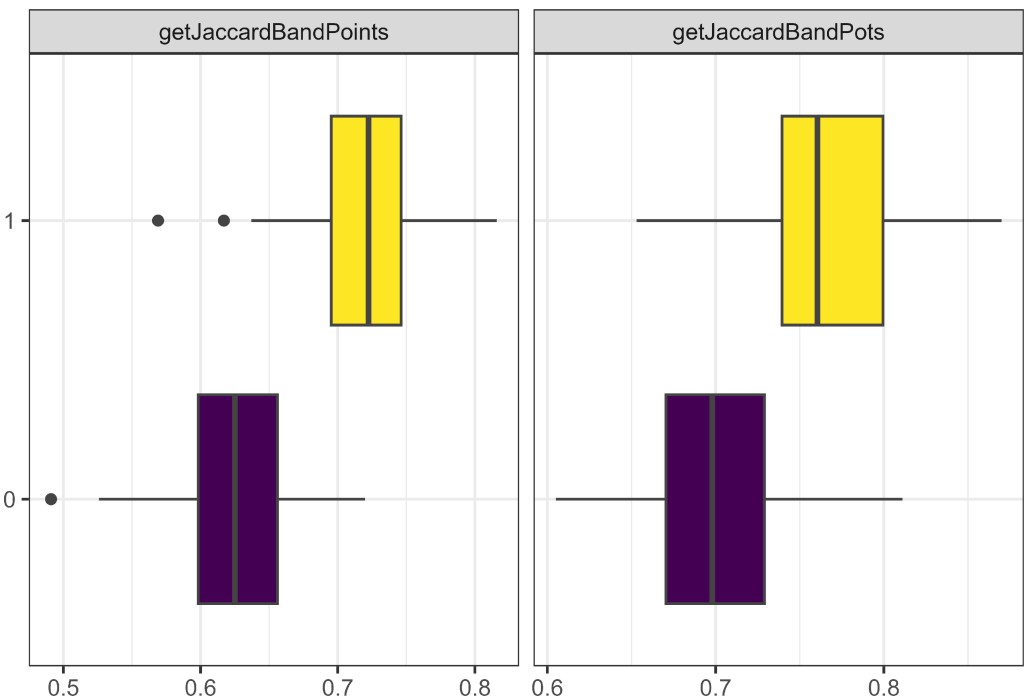

**Figure 7** **Boxplot demonstrating the correlation between the gendered migration parameter and the Jaccard distances.** The x axis is the Jaccard distance between bands' point and pot assemblages. The y axis is whether gendered migration is on (1) or off (0).

traits are expected to spread more easily, thus decreasing distance between bands. The fact this did not happen with pots appears to be caused by the lack of any selective pressure. Visibility had a strong, negative correlation with the frequency of learning. There are fewer low-visibility traits, which means that if a low-visibility trait is chosen for transmission then occasionally nothing is transmitted.

Pottery prestige is either assigned randomly at the start of the model or one person is chosen at random as the prestigious individual. Neither method produced any meaningful correlations.

## DISCUSSION

A necessary step prior to using the ArchMatNet agent-based model to draw inferences about the ability of researchers to reconstruct ancient social networks using material culture is to understand how the modeled social, cultural and environmental processes interact with one another and affect the simulated material culture. The results of the sensitivity analyses presented above shed light on these matters, and we hope can be used as reference for both characterizing and interpreting the model when applying it to different case scenarios.

One of the first things that becomes evident from the sensitivity analyses, as expected, is that given the stochastic nature of agent-based models, there is variability in the effect of different parameters on model outputs. Because of that, several runs of the model should

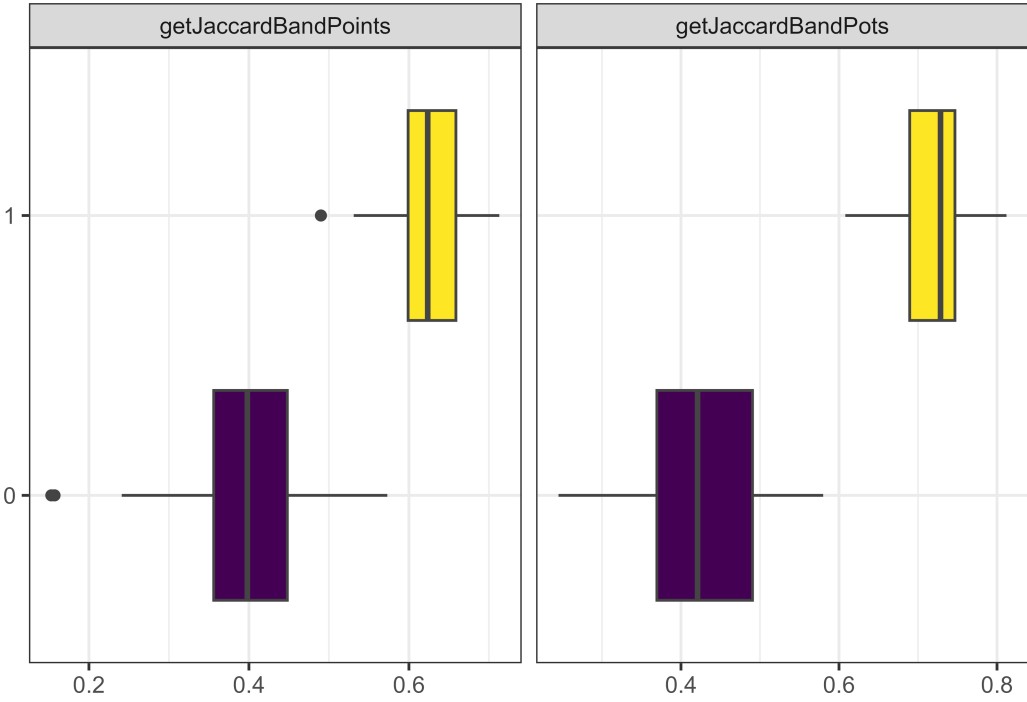

**Figure 8** **Boxplot demonstrating the correlation between the unique traits parameter and the Jaccard distance metrics for bands and points.** The x axis is the Jaccard distance between bands' point and pot assemblages. On the y axis, one means the parameter is true and zero is false.

always be performed for each of the combination of parameter values of interest. We also found that running the model for longer than 2000-3000 time steps did not significantly alter its outputs for most metrics–although the selective pressures on points produced some continuous change, therefore, unless researchers are interested in a particular period of time that they wish to recreate, the relationships between the model dynamics and its outcomes do not require running it for longer.

The frequency of environmental change did not significantly affect the frequency of interactions between agents for any of the activities, and it also did not influence any of the cultural outcomes of the model, including the size of assemblages nor the overall cultural differentiation. This implies that environmental instability does not seem to mediate the relationship between social and archaeological networks. Note that this does not refer to particular environmental characteristics of environments (*e.g.,* soil pH, precipitation patterns, humidity...), which in real settings are major determinants of the preservation of archaeological material as well as well-known drivers of geographical and temporal biases in the archaeological record (*Friesem & Lavi, 2017*; *Gravel-Miguel et al., 2022*).

As expected, larger populations and more frequent visits resulted in larger assemblages, as more agents had the opportunity to produce and obtain objects. Larger camp populations also resulted in less differentiated assemblages. Although at first glance these results might seem counter-intuitive, the larger a camp population is, the less likely that traits are lost completely by chance. Therefore, given sufficient levels of interconnectivity, a larger

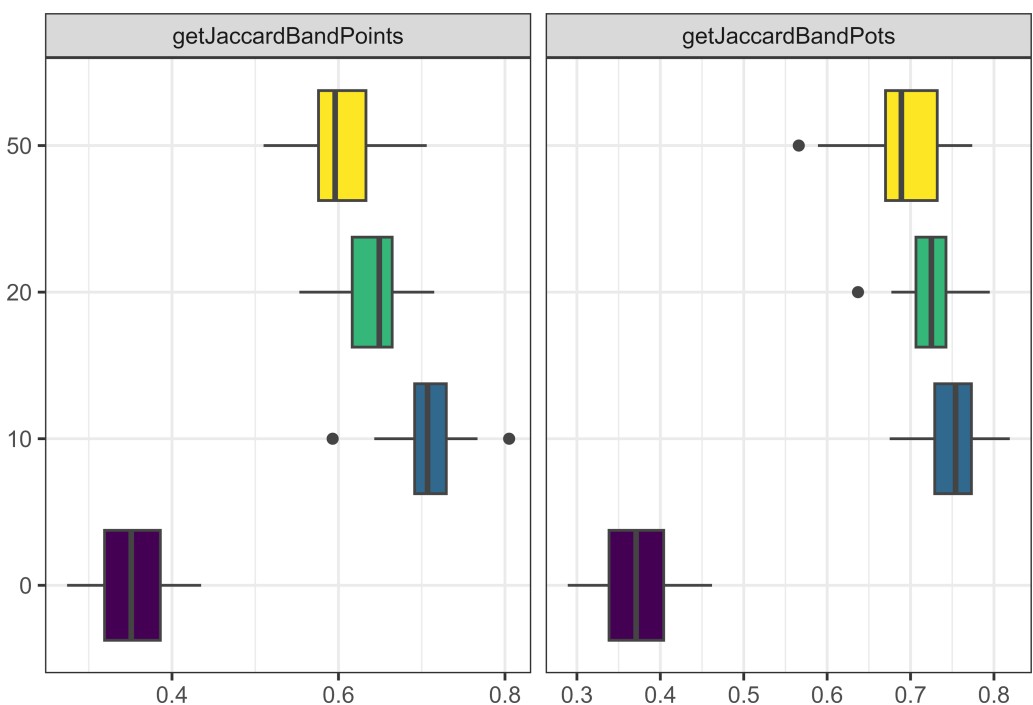

**Figure 9** **Boxplot demonstrating the correlation between the probability of loss parameter and the Jaccard distance metrics for bands and points.** The x axis is the Jaccard distance between bands' point and pot assemblages. The y axis indicates the probability of losing a trait in percentages.

population is able maintain the traits that have been acquired through inter-camp or inter-band interactions. As other studies have also postulated, migration was an important factor for ensuring cultural exchange between camps, and bands (*Creanza, Kolodny & Feldman, 2017*; *Derex, Perreault & Boyd, 2018*). Hence, it reduced the cultural differentiation between both pot and point assemblages. We also found gendered migration to have an effect on the cultural dynamics of the model by increasing the cultural differentiation between assemblages. As has been previously hypothesised, the number of agents exchanging traits with non-local individuals is reduced, when only one gender is allowed to migrate/disperse (*Dyble, 2018*). Therefore, even when migration rates were equal, the non-migrating gender is constrained in their ability to learn and transmit cultural traits.

The probability of hunting or visiting resulted in less distinct pot assemblages between camps. Individuals were allowed to trade or learn during hunting trips. Therefore, given that pot exchange or learning, unlike point exchange and learning did not depend on hunting success, it is perhaps easier for individuals to acquire pots or pot traits from others from different camps during such hunting trips leading to a greater similarity in camp assemblages.

It is important to note that we modeled as an adjustable parameter the rate at which agents drop objects in assemblages. Nonetheless, we did not model the preservation or recoverability of archaeological material as stochastic processes. This means that in real settings, it is very plausible to assume that, in reality, the effect of assemblage size on our

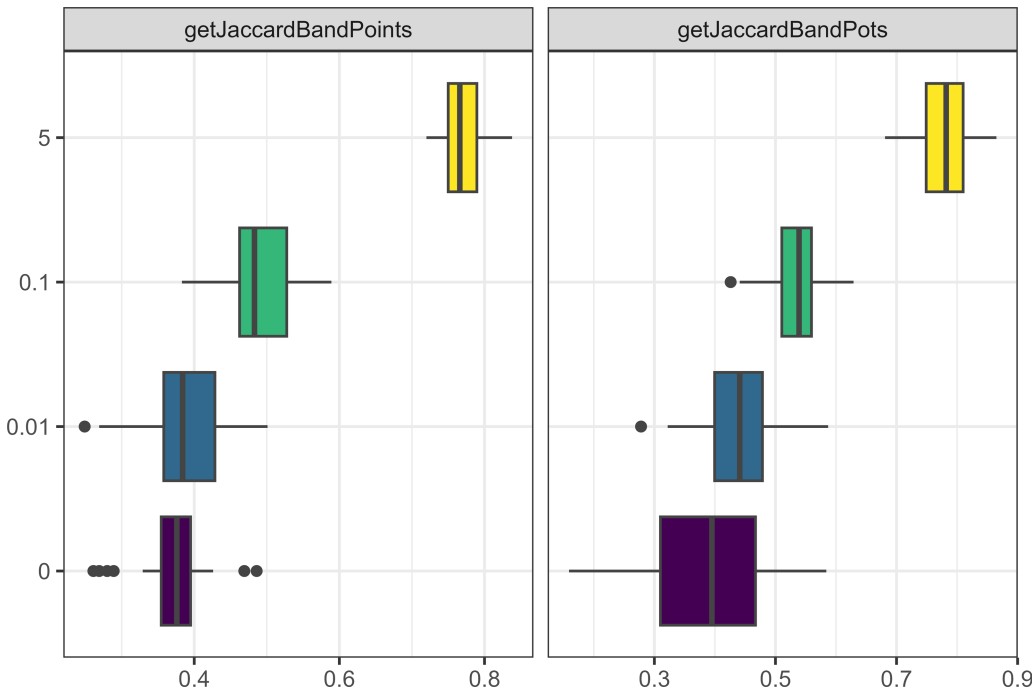

**Figure 10 Boxplot demonstrating the correlation between the unique traits parameter and the Jaccard distance metrics for bands and points.** The x axis is the Jaccard distance between bands' point and pot assemblages. On the y axis, one means the parameter is true and zero is false.

ability to recover ancient social networks from archaeological networks might be much stronger than observed in our model as those factors that lead to larger assemblages will also increase the probability that some archaeological material is recovered from them.

Of all the cultural factors, starting with homogeneous or heterogeneous material culture was the most impactful on the primary metric to measure cultural difference–Jaccard distance between bands. Also impactful were parameters controlling the rates of innovation and the frequency and types of cultural transmission: prestige *versus* conformism. Closely related to transmission is the possibility of losing a trait, which also had a noticeable impact on cultural distance. The visibility of a trait also had a major impact on the distance between traits in bands, although this relationship differed between pots and points. The probability of creating a new object and the types of pottery prestige of an object had little effect on the Jaccard distance.

Modeling cultural transmission has been done in several ways. Graham (*Graham, 2006*) used a simple contagion model for understanding space in Rome. Carrignon, Brughmans, and Romanowska (*Carrignon, Brughmans & Romanowska, 2020*) used randomly sampled rates of innovation, learning, and interactions for comparing inter-regional Roman trade. Brughmans and Poblome (*Brughmans & Poblome, 2016*) used the presence or absence of links to determine transmission of objects in a model of the Roman economy. Premo (*Premo, 2014*) used a cultural transmission model to test diversity in assemblages by randomly transmitting traits from generation to generation while introducing copying

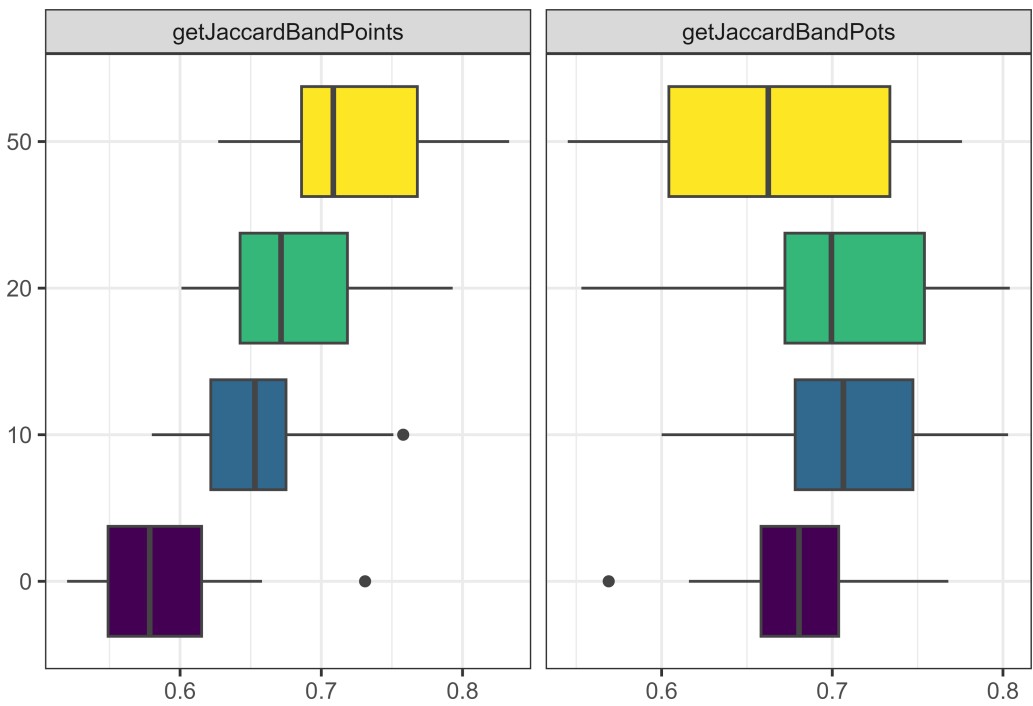

**Figure 11  Boxplot demonstrating the correlation between the transmission rate parameter and the Jaccard distance metrics for bands and points.** One means the parameter is true and zero is false.

errors. Early experiments with numeric traits and copying error were unable to produce stylistic variation between camps (*Padilla-Iglesias, Bischoff & Gravel-Miguel, 2021*). Hence, our approach combined the conformist and prestige transmission used by Eerkens and Lipo (*Eerkens & Lipo, 2005*) with a main tool and variant approach inspired by Kolodny, Creanza, and Feldman's work (*Kolodny, Creanza & Feldman, 2015*; *Creanza, Kolodny & Feldman, 2017*). This allowed the creation of stylistic variation between camps, even if it takes some time to develop as well as the generation of change over time without the need of introducing generational change. This is due to the loss of traits over time. We included multiple avenues and types of transmission in order to represent cultural transmission as accurately as we could within reasonable constraints. This in turn generates more realistic social networks and archaeological assemblages, the comparison of which is the ultimate goal of this model. We have demonstrated that the parameters in this model produce meaningful variation between bands and generate a variety of different types of interaction.

This description and sensitivity analysis of the ArchMatNet model evaluates the choices made in developing the model in preparation for exploring the questions it was designed to answer. Although the model is not meant to be a life-like reconstruction of prehistoric behavior, we have attempted to balance the complexity that allows for the testing of numerous conditions pertinent to real life scenarios with simple behaviors that allow for interpretability. The model is available for reuse, and we believe it can be adapted for a variety of purposes beyond our original intentions.

## ACKNOWLEDGEMENTS

We gratefully acknowledge the co-creator of ArchMatNet, Claudine Gravel-Miguel, for her substantial contributions to the development of the model and for her many other contributions to this project. We also recognize Arizona State University's Research Computing for the use of their high computing cluster to run the analysis.

### Funding

The authors received no funding for this work.

### Competing Interests

The authors declare there are no competing interests.

### Author Contributions

- Robert J. Bischoff conceived and designed the experiments, performed the experiments, analyzed the data, performed the computation work, prepared figures and/or tables, authored or reviewed drafts of the article, and approved the final draft.
- Cecilia Padilla-Iglesias conceived and designed the experiments, performed the experiments, analyzed the data, performed the computation work, authored or reviewed drafts of the article, and approved the final draft.

### Data Availability

The results of the model runs, the R script used to produce this document, and the full sensitivity analysis are available at the Open Science Foundation: Bischoff, Robert, Cecilia Padilla-Iglesias, and Claudine Gravel-Miguel. 2023. "Supplemental Material for A Description and Sensitivity Analysis of the ArchMatNet Agent-Based Model." OSF. April 11. doi: 10.17605/OSF.IO/C6M5T.

The ArchMatNet model has been peer reviewed through CoMSES Net (the Network for Computational Modeling in Social and Ecological Sciences) and can be found here along with the ODD (description of the model):

https://www.comses.net/codebases/c481ae31-0c2f-49ab-8430-57945a4b6213/releases/1.1.0/.

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
