# Peer review of "A description and sensitivity analysis of the ArchMatNet agent-based model"

_PeerJ Computer Science, doi:10.7717/peerj-cs.1419_

## Round 0.1 · original submission · Minor Revisions

Thank you for submitting your manuscript to our journal. I have carefully reviewed the comments provided by our two reviewers, and I am pleased to inform you that your submission has been assessed as requiring minor revisions.
Below, I have summarized some of the key points raised by the reviewers, which you may consider addressing in your revision to improve the article further. The reviewers already find the current manuscript to be of merit and relevance.

The reviewers suggest completing the literature in the introduction, background, and certain parts of the text with examples where applicable. There are also suggestions related to the legibility of the figures. An interesting aspect is providing the data in more standard formats so that a wider range of software can be used. Additionally, there are suggestions regarding the description of both the model and the methodological pipeline of the work.

Please ensure that you try to address these points and other minor suggestions to improve your revised manuscript. Once you have made the necessary revisions, please resubmit your manuscript for further consideration.

Thank you for considering our journal as a venue for your work. We look forward to receiving your revised manuscript.

Sincerely

·

Basic reporting

No comment.

Experimental design

No comment.

Validity of the findings

No comment.

Additional comments

I don't know exactly where to fit these comments into the boxes above, so I'm putting them all here. They touch variously on the criteria outlined above (e.g., Basic Reporting, Experimental Design, and Validity of the Findings).

First off, this is such an intriguing tool that the authors have built. I have so many ideas about how I can use this in my own work. It is an excellent platform that will certainly contribute to enhancing network applications in archaeology and beyond. I have only a few comments that I think the authors needs to address.

First, in the intro, the authors state "The ArchMatNet agent-based model (ABM) was designed to test archaeologists’ ability to reconstruct prehistoric social networks from networks of material culture under different conditions." This isn't quite what the ABM is doing. There is not really any "testing" of archaeological results. There is simply model building. The output of the ABM can be productively comapred to archaeological data, which can produce hypotheses about where the model and the observations diverge or converge. But the validity of the archaeological data is not being tested. The ABM does not produce an indisputable baseline of behavior uopn which to test how well the archaeological record caputres those behvaior. Human interactions are much too complex. Rather, the ABM gives an informed blueprint (an idealized model built from a limited set of parameters of human behavior) against which to interpret archaeological datasets. Which is great! But it isn't really a "test."

In the intro, the authors need to be clear about that kinds of societies are being modeled, what scales, what kinds of activities, behaviors, etc. This isn't a model that can be applied to any society anywhere or any behaviors. We're just missing a description of the kinds of cases this ABM is actually modeling.

How to account for gendered production? What if only women make pots? What if only men make points? For example, if men and women both migrate, transmission related to pottery production may equal that of a scenario where only women migrate. But if only men migrate, transmission related to pottery may remain low, while transmission of point styles increases. Right now, there is no way to assign a gender to a material (that I can see).

Similarly, it is rare that ALL individuals in society create pots or points that would enter circulation (e.g., children v. adults, married v. unmarried women, members of a single lineage, etc.). By not accounting for the heterogeneity of (1) those who produce what and thus (2) those who may transmit or discard, the model isn't really capturing the dynamic relationships between interaction and material culture production.

Finally, I would really like to see the authors apply the model to an ethnographic case from the literature on costly signaling, transmission, or something of the sort. What happens when you use parameters from a known ethnographic case? How will the model outputs compare to the known case? I know this maybe hard to test against material networks, as that info isn’t commonly recorded in ethnography, but the modeled social networks of person-to-person or camp-to-camp could be compared against known interaction behaviors/patterns in the ethnographic record (even where it isn't neceaarily network data). We need to see that the ABM can replicate observed patterns of interaction before we use it to generate hypotheses about past interactions and the production of the archaeological record.

Cite this review as

·

Basic reporting

The article is written in a clear and unambiguous manner, conforming to all professional standards.
The article generally includes sufficient introduction and background to link it to the broader field of knowledge. However, I was surprised to see some highly relevant recent work not ebing cited or engaged with, most notably Kanters et al. recent paper on sensitivity analysis in archaeological simulations (Kanters, H., Brughmans, T., & Romanowska, I. (2021). Sensitivity analysis in archaeological simulation: An application to the MERCURY model. Journal of Archaeological Science: Reports, 38, 102974. https://doi.org/10.1016/j.jasrep.2021.102974). Including works such as this one will have a clear additional value for this paper.
Certain points could be backed up by additional citations as well. E.g. line 121-122: "This number is somewhat arbitrary, but it is based on expectations derived from a broad reading of
ethnographic literature" --> cite some examples from the literature.
The structure of the paper does not strictly conform to the standard sections requested in the Instructions for Authors. However, the structure applied here is logical and clear within the context of the paper itself, so I don't have any issues with it.
All figures are relevant and most are legible. Only figure 1 is a bit crowded and not all connections completely align (for example below the "go" box). I would consider simplifying this if at all possible. In figure 3, you could perhaps slightly reduce the font size for the variables as it looks a bit cramped now.
The raw data is available through the .Rds files and R script found in the OSF repository, but not as CSV which would allow people to access the data regardless of the software they use. I would recommend adding this.
The submission is mostly self-contained, however, given that it pertains to a sensitivity analysis of an existing model, the basics of certain aspects of the model are not explained to the fullest or not discussed at the right moment within the overall flow of the paper.
E.g.:
line 108: "Objects from those inventories are created based on known traits." --> What do you mean with known traits?
Line 111-112: "Each point trait is adaptive in the environment where it was invented" --> What do you mean by where it was invented?
Line 127: "People remove any superfluous alliances they may have accumulated in the previous time step" --> at this point you have not yet mentioned anything about alliances, let alone ones that might be superfluous.
I think all of these example stem from the fact that the paper does not contain a full description of the model and how it works. This is not necessarily a major issue as you also provide the ODD as part of the repository, but I would go through the entire model description part of the paper and make sure that you provide all necessary information in the right order so that a reader with no prior knowledge of the model knows exactly what is going on at every step of the way.

Experimental design

The submission has a clearly defined and relevant research question. Sensitivity analysis is an important aspect in the process of conducting experiments and simulations, yet has only recently started to be explicitly acknowledged in the archaeological literature.
The tests performed are rigorous and adhere to all technical standards as far as I'm aware of them. All steps of the process can be sufficiently reconstructed from the associated R script. However, you might want to describe and explain the methodological pipeline in some more detail in the paper itself as well. As it stands, the reader is sometimes left to put the puzzle together for themselves as to how the different steps of the process come together.

Validity of the findings

The findings are well stated. The paper is well set up to encourage replication. A potential addition could be to include a short section where you offer the reader an explicit pipeline for conducting sensitivity analysis themselves as well. I always like Matt Peeples' tutorials as good examples of this: Peeples, Matthew A. 2017 Network Science and Statistical Techniques for Dealing with Uncertainties in Archaeological Datasets. [online]. Available: http://www.mattpeeples.net/netstats.html.
Note that it's just a suggestion and you obviously don't have to be extensive as in this example. Just providing a short recap/overview of all the necessary steps could already offer a great added value.

Additional comments

I don't really have any major comments. All of the preceding are suggestions for how you could make an already good paper even better (in my humble opinion of course). I enjoyed reading the paper. Thank you for this important contribution!

---

## Round 0.2 · accepted · Accept

Dear authors,

I am pleased to inform you that your manuscript titled "A description and sensitivity analysis of the ArchMatNet agent-based model" has been accepted for publication in PeerJ Computer Science following the revisions you have made based on the feedback provided by the reviewers.

The changes you have made have improved the quality and clarity of the manuscript, and we believe it will make a valuable contribution to the field. We appreciate the effort you have put in to address the reviewers' comments and suggestions, and we are excited to publish your work.

Thank you for submitting your work to our journal, and we look forward to publishing more of your research in the future.

Best regards,
José Manuel Galán